# Load-Balancing Routing for LEO Satellite Network with Distributed Hops-Based Back-Pressure Strategy

**DOI:** 10.3390/s23249789

**Published:** 2023-12-12

**Authors:** Chi Han, Wei Xiong, Ronghuan Yu

**Affiliations:** 1Science and Technology on Complex Electronic System Simulation Laboratory, Space Engineering University, Beijing 101400, China; 15850466132@163.com (C.H.); yrh1983@163.com (R.Y.); 2School of Space Information, Space Engineering University, Beijing 101400, China

**Keywords:** load-balancing, back-pressure routing, LEO satellite network, routing protocol, hops-count

## Abstract

With the expansion of user scale in LEO satellite networks, unbalanced regional load and bursty network traffic lead to the problem of load disequilibrium. A distributed hops-based back-pressure (DHBP) routing is proposed. DHBP theoretically derives a fast solution for the minimum end-to-end propagation hops between satellite nodes in inclined-orbit LEO satellite networks; hence, link weights are determined based on remaining hops between the next hop and destination satellites. In order to control the number of available retransmission paths, the permitted propagation region is restricted to a rectangular region consisting of source-destination nodes to reduce the propagation cost. Finally, DHBP is designed distributedly, to realize a dynamic selection of the shortest link with low congestion and balanced traffic distribution without obtaining the whole network topology. Network simulation results demonstrate that DHBP has higher throughput and lower delay under high load conditions compared with state-of-the-art routing protocols.

## 1. Introduction

The LEO satellite network (LSN) has the characteristics of wide coverage as well as a large communication capacity, which breaks through geographical restrictions by means of inter-satellite networking. Based on the above advantages, the LSN can provide global users with a high transmission rate, low delay, and seamless network access services. At present, the LEO satellite network plays an important role in the space–air–ground integrated network (SAGIN). The orbit altitude of the LEO satellite network ranges from 160 km to 2000 km. Compared with satellites deployed in geostationary orbit (GEO), the LSN obtains a better transmission rate and lower delay under less energy consumption [1,2,3]. For example, the polar-orbiting constellation, OneWeb, consists of 720 satellites with an orbital altitude of 1200 km. OneWeb is able to provide Internet connectivity around the world [4]. The Starlink constellation was proposed by SpaceX in 2015 [5]. Starlink plans to deploy about 42,000 satellites to create a giant LEO constellation to achieve near-global broadband Internet access.

Due to the sudden time of network traffic and the uneven geographical distribution of terrestrial users, with the continuous expansion of user scale, the problem of load imbalance in LEO satellite networks is becoming more and more prominent. The LEO satellite network integrated with the terrestrial network is shown in Figure 1. The communication between two ground gateway stations is usually busy, which leads to congestion of the shortest path between them during the communication peak. In this case, the giant mesh topology of LSN provides multiple candidate paths. Through reasonable management and distribution of network load, a high transmission rate as well as low delay can be taken into account.

Aiming at the load-imbalancing of the LSN, several load-balancing routing algorithms for the LSN have been proposed in recent years. Dong et al. [6] proposed a load-balancing routing algorithm for 6G SAGIN based on Deep Q-Network. Li et al. [7] proposed a selective iterative Dijkstra algorithm (SIDA) to optimize the path finding process by reducing the reuse frequency of nodes. A selective split load-balancing strategy (SSLB) is proposed to solve the link congestion problem in LSNs. Li et al. [8] put forward a state-aware load-balancing (SALB) strategy, which estimates the link status and dynamically sets the queue delay weight. By adopting the shortest path tree to reduce the overhead, the throughput of LSN is optimized. Wang et al. [9] put forward load-balancing routing based on congestion prediction (LBRA-CP), which converts the load-balancing problem to a multiple-target optimization problem. By predicting link congestion, intelligent optimization algorithms are utilized to solve the optimal path and improve the throughput of LSN. Liu et al. [10] proposes the low-complexity routing (LCRA), which utilizes the congestion information of adjacent nodes to select the next hop with low congestion as the forwarding direction. When the path is unique, LCRA chooses to wait for a while before forwarding the packet, which improves the network transmission rate and reduces delay. The above research mostly depends on the global link-state information [11,12], which leads to poor effect under heavy load.

Back-pressure (BP) routing processes the routed traffic through the congestion gradient of the packet queue between nodes, which performs excellent equalization effect in multi-hop networks, such as wireless sensor networks (WSNs) and mobile ad-hoc networks (MANETs) [13,14]. Similar to multi-hop networks, data packets in the LSN follow the same characteristics. Therefore, BP routing is also suitable for the traffic burst and load imbalance of LSN. However, compared with WSNs or MANETs, nodal distance in the LSN is larger. The network topology is also consistent with the Manhattan network [9]. As a result of the above properties, there are many end-to-end alternative paths in the LSN. In this case, the original back-pressure routing will search for ample end-to-end paths, resulting in huge end-to-end delay and computation complexity. Based on the above analysis, the original BP routing is not applicable in LSN.

In this paper, end-to-end hops count in the LSN, as a key transmission cost metric is introduced into BP routing. Different from traditional WSNs, inter-node distance in the LSN can reach hundreds or even thousands of kilometers. Take Starlink constellation Shell1 as an example [3]; when the orbit height is 550 km, the distance between adjacent satellites in the same orbit is about 1972 km. For neighboring satellites in adjacent orbits, the distance varies from 880 km to 1400 km. Due to the huge distance between neighboring nodes, end-to-end transmission hops play an important role in network delay. During the optimization of the propagation path, the end-to-end residual hop count should also be considered on the basis of the BP strategy. This paper proposes a distributed hops-based BP routing (DHBP) and cache scheduling scheme, which combines the BP strategy with the end-to-end hops count to realize adaptive link load aware routing. At the same time, the streaming limit domain is introduced to reduce the propagation delay. Simulation results demonstrate that the proposed DHBP is able to optimize the network performance under the condition of lower transmission cost when compared with the centralized strategy [8,9].

The main contributions of this work are summarized as follows:Increased constellation scale leads to excessive complexity in interstellar hops count estimation. In response to this problem, based on analyzing the characteristics of inclined constellations, a theoretical model is proposed to explicitly estimate the minimum end-to-end hops count and the corresponding propagation direction, so as to reduce the computational complexity;Considering the heavy load in the LEO satellite network, we use the destination-hops-delay (DHD), i.e., calculate the transmission delay according to the estimated end-to-end hops count from the next hop to destination nodes. The DHBP for the LSN is proposed. DHBP measures the backlog of BP routes by DHD and dynamically selects the shortest path with low congestion to balance the traffic overhead;The network stability of the DHBP is analyzed by utilizing the time-delay measurement based on hops count, and the throughput optimization is proved in the LSN. Network simulation is conducted on OMNET++ to test the performance of the proposed scheme. Simulation results demonstrate that DHBP optimizes throughput and transmission cost, especially under bursty traffic environments.

In Section 2, we summarize two ideas for load-balanced routing and analyze the current state of the research on BP routing. In Section 3, we establish the system model of the LEO satellite network. The estimation method of end-to-end hop count in the LSN is deduced. In Section 4, we introduce the DHBP routing. In Section 5, we analyze the network stability of DHBP, as well as throughput optimization. In Section 6, the simulation and the results are analyzed, and the conclusions are presented in Section 7.

## 2. Related Works

In the early stage of the development of the LSN, due to the low traffic load, researchers assumed that communication delay was mainly determined by propagation delay. Most of the routing strategies focused on minimizing end-to-end propagation delay. However, with the increase of network traffic load, queuing delay in satellites becomes an important part of communication delay. An effective routing policy must prevent traffic from passing through congested nodes.

Load-balancing routing can be classified into global information-based and local information-based routing. The former mainly includes satellite networks link state routing (SLSR) [14], agent-based load-balancing routing (ALBR) [15], a load-balancing routing algorithm based on congestion prediction (LBRA-CP) [9], and state-aware and load-balanced routing (SALB) [8], etc. SLSR, ALBR, LBRA-CP, and SALB all belong to single-path traffic balancing schemes, which can only allocate traffic to other paths with a lower link cost when the traffic is heavy, but cannot fundamentally eliminate congestion. Tang et al. [16] proposed network coding-based multi-path cooperative routing (NCMCR), which employs multi-path cooperative routing based on network coding so that different parts of data flow are delivered along multiple disjoint paths. Traffic on each path is allocated based on path queuing delays. Mohorcic et al. [17] proposed two forwarding strategies based on alternate link routing (ALR) to divert traffic from the shortest path to the alternate shortest path. Different from ALR, compact explicit multi-path routing (CEMR) [18] uses an orbital speaker mechanism to periodically collect and exchange the link state information of the whole network, and calculates *k* shortest paths for data transmission. Through the PathID coding scheme, the balance of traffic load in the satellite network is supported with a lower signal overhead. Although ALR and CEMR utilize multiple paths for traffic distribution, the traffic on each path is allocated randomly. The congestion on the diversion link may be exacerbated after traffic distribution. Since the state of other links is not fully considered, it is difficult to achieve the desired equalization effect.

Different from the global strategy, local load-balancing makes routing decisions only based on local information, mainly including Priority-based Adaptive Routing (PAR) [19], Explicit Load-Balancing (ELB) [20], Traffic-Light-Based Intelligent Routing (TLR) [21], etc. Most of the above load-balancing algorithms carry out the transmission rate or link adjustment after the onset of congestion. To address this problem, researchers proposed load balanced routing based on congestion prediction [22], including Hybrid Global-Local load-balancing Routing (HGL) [23], etc. HGL decomposes satellite network traffic into a predictable large-scale baseline and a series of unpredictable small-scale random fluctuations. Aiming at the large-scale traffic baseline, HGL adopts a global balancing strategy to actively distribute traffic across the whole network. After that, in order to deal with the sudden congestion caused by small-scale random fluctuations, HGL uses a local balancing strategy to quickly adjust routes and alleviate congestion. Therefore, there are limitations in both global and local strategies. In this work, a low cost and scalable distributed routing scheme is designed to enhance the load-balancing ability of the LSN. Liu et al. [24] proposed a low computation complexity routing algorithm (LCRA). LCRA selects the less congested next hop according to the congestion information of neighboring satellites, which improved delay and data delivery ratio.

BP routing, which adopts the congestion gradient of packet queues between nodes to deal with dynamic routing traffic and stabilizes the random network in the process of routing and resource allocation, is widely used in multi-hop networks. Hai et al. [25] proposed an improved BP algorithm called sojourn-time-based back-pressure routing (STBP) by introducing the sojourn time backlog (STB). STB takes queue length and cumulative packet delay into account, which improves the delay performance of BP. Ying et al. [26] proposed the shortest-path-aided BP routing (SBR) by introducing short-path information. Nevertheless, the scalability of SBR is limited due to centralized strategy. Wang et al. [27] proposed differential BP routing (DBPR) with single queue management, which simplifies data queue management and improves packet delay performance. Zhao et al. [28] proposed a throughput-optimal biased BP algorithm (OBBP). The bias is learned through GNN, which seeks to minimize end-to-end delay. Chen et al. [29] developed a throughput-optimal scheme utilizing the utility maximization framework for data collection through satellite networks. The shortest path is determined by the distance between adjacent nodes and sink nodes. A comparison of load-balancing routing algorithms is presented in Table 1.

Due to the regular topology of the LSN, a large number of available paths may lead to large delays and overhead. The above strategies based on BP routing cannot be adopted to the LSN directly. In this work, we introduce end-to-end hops in the LSN into weight determination, and transmission is also restricted to a rectangular region, which is different from other studies.

## 3. System Model

### 3.1. LEO Satellite Network

The topology of the LSN can be demonstrated by G=(V,E), where *V* denotes satellite nodes and *E* denotes inter-satellite links (ISLs) between satellites. Inter-satellite links include intra-plane ISLs and inter-plane ISLs. The former are used to connect front and rear satellites in the same orbit, while the latter are adopted to connect adjacent satellites in adjacent orbits. Assume that the number of satellites in the LSN is *N*, and each satellite contains four ISLs including two intra-plane ISLs and two inter-plane ISLs, as shown in Figure 2.

Due to the mesh topology of the LSN, it can be regarded as a Manhattan network [32]. Figure 3 demonstrates the 2D topology of an LSN with *n* orbits, where *m* satellites are distributed evenly. Each node contains two intra-plane ISLs (blue lines) and two inter-plane ISLs (black lines). The motion of satellites in orbit is periodic and predictable [33]. It is assumed that, over a small period of time, the LSN topology remains constant [34]. Obviously, the length of intra-plane ISLs is relatively stable. Inter-plane ISLs in low latitudes have a longer length while inter-node distances in high latitudes are relatively shorter.

### 3.2. End-to-End Hops in LSN

In the LSN, there is a large distance and propagation delay between satellite nodes. The remaining hops to the destination can be used to make packets propagate along a shorter path to improve network delay performance [35,36]. In this subsection, we will analyze the shortest propagation hops between any nodes in the inclined orbit constellation, so as to provide a basis for the improvement of the back-pressure routing strategy.

Given a Walker-delta constellation composed of m×n satellites, as shown in Figure 3, the dip angles of all orbits are α. The ascending points of all orbits are uniformly distributed along the equator. The right ascension of the ascending node (RAAN) of the adjacent tracks is ΔΩ=2π/n. The phase difference between adjacent satellites on the same orbit is ΔΦ=2π/m. Given that the phase factor of the constellation is *F*, the phase difference of the adjacent satellites in adjacent orbits is Δf=2πF/mn. Figure 4 demonstrates the trajectory of the substellar points of the LSN. The hops count required for the path from source to destination satellites (purple arrow) consists of two parts: inter-plane hops $H_{h}$ in the horizontal direction and intra-plane hops $H_{v}$ in the vertical direction.

#### 3.2.1. Inter-Plane Hops $H_{h}$

The hops moving laterally between inter-planes are determined by the longitude difference $\Delta L_{0}$ of the ascending point of the initial orbit. Given the source and destination satellites, $Sat_{1}$ and $Sat_{2}$, as shown in Figure 4,
(1)$$\Delta L_{0}=\left(L_{2}-L_{1}\right)\;\;\mathrm{mod}\;\;2\pi \in \left[0,2\pi \right],$$
where $L_{1}$ and $L_{2}$ denote the ascending meridians of $Sat_{1}$ and $Sat_{2}$, respectively. If the destination satellite is on the west side, the longitude difference is $2\pi -\Delta L_{0}$. Since the longitude difference of the ascending points of the adjacent orbits is constant, i.e., ΔΩ=2π/n, hops moving in the east and west directions can be, respectively, expressed by: (2)$$H_{h}^{\leftarrow }=\langle \frac{2\pi -\Delta L_{0}}{\Delta \Omega }\rangle ,H_{h}^{\to }=\langle \left|\frac{\Delta L_{0}}{\Delta \Omega }\right|\rangle ,$$
where $\langle x\rangle =\mathrm{sgn}\left(x\right)\left\lfloor x+0.5\right\rfloor$ indicates that *x* is rounded to the nearest integer. $H_{h}^{\leftarrow }$ is the hops count between westward orbits and $H_{h}^{\to }$ is the hops count between eastward orbits.

#### 3.2.2. Intra-Plane Hops $H_{v}$

According to the above analysis, inter-plane hops is determined by the longitude difference $\Delta L_{0}$ between the ascending points of source-destination satellites. Correspondingly, intra-plane hops depend on the phase angle difference Δu between satellites. Since each intra-plane hop increases the phase angle by ΔΦ and each inter-plane hop increases the phase angle by Δf, the phase angle of the destination satellite $Sat_{2}$ can be expressed as: (3)$$u_{2}=u_{1}+\left(H_{h}^{\to }\cdot \Delta f\right)+\underset{\Delta \vec{u}}{\underset{︸}{\left(H_{v}^{↗}\cdot \Delta \Phi \right)}},$$
where $H_{h}^{\to }$ is the eastward inter-plane hops count and $H_{v}^{↗}$ is the eastward intra-plane hops count. In order to calculate intra-plane hops $H_{v}$, the phase difference $H_{h}^{\to }\cdot \Delta f$, resulting from $H_{h}$, needs to be eliminated from overall phase difference $\Delta u=u_{2}-u_{1}$. Similar to the calculation of the inter-plane hops count, the calculation of phase difference Δu distinguishes between east and west propagation directions can, respectively, be expressed by: (4)$$\left\{\begin{array}{c} \Delta \vec{u}=\left(u_{2}-u_{1}-H_{h}^{\to }\cdot \Delta f\right)\;\;\mathrm{mod}\;\;2\pi \\ \Delta \overset{\leftarrow }{\;u}=\left(u_{2}-u_{1}+H_{h}^{\leftarrow }\cdot \Delta f\right)\;\;\mathrm{mod}\;\;2\pi , \end{array}\right.$$
where Δf denotes the change of phase angle resulting from inter-plane hops. $H_{h}^{\to }$ denotes the westward hops count and $H_{h}^{\leftarrow }$ denotes the eastward hops count.

Satellite orbits include ascending orbits (from southwest to northeast) and descending orbits (from northwest to southeast). The ascending and descending orbits can also be divided into forward hop and backward hop. In this work, we calculate inter-plane hops in four directions: (5)$$\left\{\begin{array}{c} H_{v}^{↖}=\left|\frac{\Delta \overset{\leftarrow }{\;u}}{\Delta \Phi }\right|,H_{v}^{↗}=\left|\frac{\Delta \vec{u}}{\Delta \Phi }\right|\\ H_{v}^{↙}=\left|\frac{2\pi -\Delta \overset{\leftarrow }{\;u}}{\Delta \Phi }\right|,H_{v}^{↘}=\left|\frac{2\pi -\Delta \vec{u}}{\Delta \Phi }\right|, \end{array}\right.$$
where ΔΦ denotes the phase difference between adjacent satellites in the same orbit; $\Delta \overset{\leftarrow }{\;u}$ denotes the phase angle difference covered by westward intra-plane hops; $\Delta \overset{\to }{\;u}$ demonstrates the phase angle difference covered by eastward intra-plane hops; $H_{v}^{↖}$ denotes intra-plane hops in the northwest direction; $H_{v}^{↗}$ denotes intra-plane hops in the northeast direction; $H_{v}^{↙}$ denotes intra-plane hops in the southwest direction; $H_{v}^{↘}$ denotes intra-plane hops in the southeast direction.

The possible $H_{h}$ and $H_{v}$ are calculated, respectively. Therefore, the minimum end-to-end hops that count *H* in the inclined constellation can be obtained by minimizing the sum of $H_{h}$ and $H_{v}$.
(6)$$H=\min\left\{\begin{array}{c} H_{h}^{\leftarrow }+H_{v}^{↖}\\ H_{h}^{\leftarrow }+H_{v}^{↙}\\ H_{h}^{\to }+H_{v}^{↗}\\ H_{h}^{\to }+H_{v}^{↘}. \end{array}\right\}$$

## 4. Hops-Based Back-Pressure Routing

In this section, we enhance BP routing by introducing the end-to-end hops count into LSN, which takes throughput and delay performance into account. A brief review of original BP routing is given before going further.

### 4.1. Queue Length-Based BP Routing

BP routing originates from the wireless packet multi-hop network proposed by Tassiulas et al. in [37], which is an algorithm for dynamic traffic allocation based on the congestion gradient between adjacent nodes [22]. BP routing does not build a specific source-destination path, but only calculates the queue backlog for each slot. The data transmission path is determined according to the criterion of queue backlog maximization [12]. Given LSN G=(V,E), (a,b) denotes the ISL between nodes *a*, *b*. At time slot *t*, $P_{a}^{c}\left(t\right)$ denotes the queue backlog at *a*, while $P_{b}^{c}\left(t\right)$ denotes the queue backlog at *b* at time slot *t*. Thus, the queue backlog difference of packet flow *c* between a,b can be denoted by $P_{a}^{c}\left(t\right)-P_{b}^{c}\left(t\right)$. Since multiple packet flows are usually converged on a single satellite, the backlog difference of the max queue on link (a,b) can be defined as $D_{ab}\left(t\right)$.
(7)$$D_{ab}\left(t\right)=\max_{c:(a,b)}\left[P_{a}^{c}\left(t\right)-P_{b}^{c}\left(t\right)\right].$$

On link (a,b), the following strategy is adopted to assign the transmission rate to packet flow *c*: (8)$$\mathrm{Maximize}:\sum_{a=1}^{N}\sum_{b=1}^{N}\mu_{ab}\left(t\right)D_{ab}\left(t\right)$$
(9)$$s.t.\mu_{ab}\left(t\right)\in \Gamma_{s(t)},$$
where $\mu_{ab}\left(t\right)$ denotes the transmission rate of link (a,b). $\Gamma_{s(t)}$ denotes permitted transmission rate matrices in the network. Based on the above strategy, BP routing is suitable for all kinds of multi-hop networks to maximize throughput, which is satisfied under any flow arrival rate and channel state probability.

### 4.2. Hops-Based BP Routing

In the above original BP routing protocols, the backlog of network nodes usually only considers the packets number in cache, rather than the packet delay [38,39]. This may cause a large delay and affect network performance. According to the above analysis, in the LEO satellite network, hops count between nodes is the key influencing factor of packet delay. In order to improve the delay performance of BP routing, the propagation delay of the packets is approximately estimated by end-to-end hops between the next hop and the destination. The corresponding delay is defined as the destination-hops-delay (DHD). Assume that $Q_{a}^{c}$ denotes the packet set of flow *c* at node *a*. For $p\in Q_{a}^{c}$, H(p) denotes the DHD from *a* to destination *b*.

In LEO satellite networks, the periodic and predictable orbits make the orbital roots easy to be determined. Therefore, according to the hops estimation method proposed above, intermediate satellites can quickly calculate the end-to-end hops count between the candidate next-hop and destinations. At node *a*, the cumulative DHD of all packets is recorded as ∑H(p). In this paper, the cumulative DHD is defined as the backlog metric ${\hat{Q}}_{a}^{c}$: (10)$${\hat{Q}}_{a}^{c}=\sum_{p\in Q_{a}^{c}}H\left(p\right)=\left|Q_{a}^{c}\left(t\right)\right|\times H\left(p\right),$$
where $\left|Q_{a}^{c}\left(t\right)\right|$ denotes the packets number in flow *c* at node *a* at time slot *t*. On this basis, at time slot *t*, the backlog difference between (a,b) based on DHD can be defined as: (11)$${{\hat{\omega }}^{c}}_{ab}\left(t\right)=\max_{c}\left[{\hat{Q}}_{a}^{c}\left(t\right)-{\hat{Q}}_{b}^{c}\left(t\right)\right].$$

The transmission rate on link (a,b) is determined according to the following strategy: (12)$$\mathrm{Maximize}:\sum_{a=1}^{N}\sum_{b=1}^{N}\mu_{ab}\left(t\right){{\hat{\omega }}^{c}}_{ab}\left(t\right)$$
(13)$$s.t.\mu_{ab}\left(t\right)\in \Gamma_{s(t)},$$
where $\mu_{ab}\left(t\right)$ denotes the transmission rate of flow *c* on link (a,b).

In this paper, the proposed routing strategy is defined as distributed hops-based BP routing (DHBP). An example of cache management and the queue backlog calculation of DHBP is shown in Figure 5. Each node maintains both the packet queue and the virtual DHD queue to determine backlog $\hat{Q}$ of each packet flow.

Given the source *s* and destination *d*, packet flow *c* is sent from *s* to *d*. The following steps are followed to determine which node is to be selected as the next hop when packet arrives at the node *A*. The queue length of flow *c* at *A* is $Q_{A}^{c}\left(t\right)=6$. Accordingly, when it comes to B,E,F, $Q_{B}^{c}\left(t\right)=4,Q_{E}^{c}\left(t\right)=4,Q_{F}^{c}\left(t\right)=5$. Therefore, the queue backlog difference from *A* to B,E,F is 2,2,1, respectively. In the standard BP routing, node *A* to node B,E have the same queue backlog difference. In the proposed DHBP, the backlog difference to the destination node is calculated according to Equation (Equation 10). In the above example, the number of hops between *B* and the destination is less than *E*. Therefore, *E* is excluded as the next hop. Assume that *B* and *F* have the same hops to the destination. In the backlog difference between *A* and *B*, *F* is $Q_{AB}^{c}\left(t\right)=2$ and $Q_{AF}^{c}\left(t\right)=1$, respectively. Since $Q_{AB}^{c}\left(t\right)>Q_{AF}^{c}\left(t\right)$, in this example, *B* will be taken as the next hop.

### 4.3. Distributed Routing Algorithm

Due to the limitations of the platform size and the power consumption of satellites in the LSN, on-board computing and storage resources are tight. The multi-hop network structure enables each satellite to connect to surrounding satellites through ISLs, which is suitable for the distributed deployment of routing protocols.

#### 4.3.1. Queue Management

Denoting the source and destination as *s*, *d*, respectively, when packet *p* arrives at the intermediate node *n* on link (a,b), the destination hop delay H(p) between the packet and destination can be calculated. When the first packet arrives at *n*, a virtual DHD queue with a backlog of ${\hat{Q}}_{n}^{c}$ is created, and H(p) is calculated. When packet *p* reaches *n*, ${\hat{Q}}_{n}^{c}$ will increase by H(p). When *p* is sent from *n* to the next hop, ${\hat{Q}}_{n}^{c}$ will decrease by H(p) correspondingly. The computational complexity of the above process is O(1).

In the LSN, assume that the number of nodes is *N*, and each node has *R* neighbors. Thus, DHBP stores a maximum of R(N−1)(N−2) virtual DHD queues. In the current LEO satellite networks, usually, R=4, as shown in Figure 3. Therefore, each satellite needs to maintain $O(N^{2})$ virtual DHD queues. According to the above analysis, the storage pressure is acceptable.

#### 4.3.2. Propagation Region Control

In the LSN, every satellite generally contains four ISLs. Therefore, the number of available paths from source to destination increases exponentially with the network scale. If the permitted propagation region is not limited, the transmission range of the packet will be too large and the delay of the network will be affected. Therefore, in this paper, the permitted propagation region is limited to the rectangular area composed of source and destination satellites, so as to reduce redundancy while ensuring adequate available links. Figure 6 demonstrates the permitted area.

The blue region denotes the permitted transmission area, while the rest is the prohibited area. DHBP allocates the link according to the defined maximum hop delay weight to transmit packets in the blue area. On the link (s,d), when flow *c* reaches *n*, packet *p* is listed at the head of transmission buffer first. Then, in the permitted propagation area, the next hop of *p* is determined according to the maximization criterion of ${{\hat{\omega }}^{c}}_{nb}\left(t\right)$. Assume that *b* is the next hop obtained by the decision.
(14)$$b=\underset{b\in N^{*}}{\mathrm{argmax}}{{\hat{\omega }}^{c}}_{nb}\left(t\right)$$
(15)$$\begin{array}{c} {{\hat{\omega }}^{c}}_{nb}\left(t\right)={\hat{Q}}_{n}^{c}\left(t\right)-{\hat{Q}}_{b}^{c}\left(t\right)=\sum_{p\in Q_{n}^{c}}H(p)-\sum_{p\in Q_{b}^{c}}H(p), \end{array}$$
where $N^{*}$ is candidate satellites set for the next hop on link (n,d). The distribution hops-based back-pressure routing (DHBP) is described in detail in Algorithm 1. Figure 7 demonstrates the flow chart of DHBP.
**Algorithm 1:**Distributed Hops-based Back-Pressure Routing
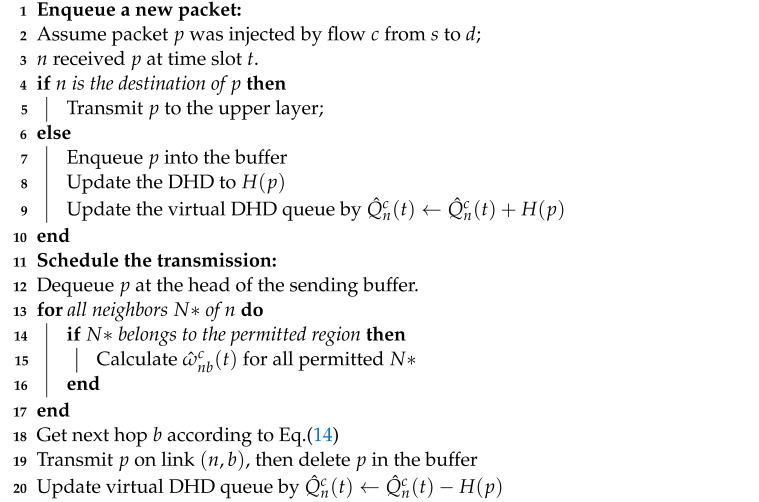


## 5. Stability Analysis

### 5.1. Network Stability of DHBP

Assume the input process as $A_{n}^{c}\left(t\right)$, and the arrival rate matrices as $\lambda_{n}^{c}$. $\lambda_{n}^{c}\left(t\right)=\lim_{t\to \infty }\left(A_{n}^{c}\left(t\right)\right)$ is the average arrival rate of $A_{n}^{c}\left(t\right)$. $\lambda_{n}^{c}$ satisfies
(16)$$P\left\{\lim_{t\to \infty }\frac{1}{t}\sum_{s=0}^{t-1}A_{n}^{c}\left(s\right)=\lambda_{n}^{c}\right\}=1.$$

The stability of the network means that, in the LSN, the total input rate is equal to the total transmission rate to the destination. Thus, the network is stable if $\lambda_{n}^{c}$ (for a long-term duration) satisfies [40]
(17)$$\lambda_{n}^{c}\leq \sum_{b\in N}\mu_{nb}^{c}-\sum_{a\in N}\mu_{an}^{c}.$$

**Remark** **1.**
*For flow c at t, define data rate $X_{a}^{c}\left(t\right)$ as the accumulation of intended routing variables:*

(18)
$$X_{a}^{c}\left(t\right)=\sum_{b\in N}\mu_{ab}^{c}\left(t\right).$$


*If $\mu_{ab}^{c}\left(t\right)$ is not scheduled, then $\mu_{ab}^{c}\left(t\right)=0$. Similarly, we also have*

(19)
$$Y_{a}^{c}\left(t\right)=\sum_{d\in N}\mu_{da}^{c}\left(t\right).$$



According to the stability requirement [40], an important direction in ensuring the stability of routing is to minimize
(20)$$\lambda_{n}^{c}-\sum_{b\in N}\mu_{nb}^{c}+\sum_{a\in N}\mu_{an}^{c}.$$

When the LSN satisfies the stability condition, the queue length should be bounded [25].

**Lemma** **1.**
*Given ${\hat{X}}_{n}^{c}=\lim_{t\to \infty }{\hat{X}}_{n}^{c}\left(t\right)$, ${\hat{Y}}_{n}^{c}=\lim_{t\to \infty }{\hat{Y}}_{n}^{c}\left(t\right)$ is the average input, output. If the LSN is stable, then*

(21)
$$\lambda_{n}^{c}-{\hat{X}}_{n}^{c}+{\hat{Y}}_{n}^{c}\leq 0.$$



The proof of Lemma 1 is given in Appendix A.

### 5.2. Throughput Optimization of DHBP

Given a queue network like the LSN, the throughput is at a maximum when the arrival rate is in a stable region. Therefore, the throughput optimization policy can be considered as a strategy that tries to ensure network stability at the upper bound of allowable $\lambda_{n}^{c}$, i.e., to minimize the DHBP difference between *t*, t+1. We define a Lyapunov drift Δt to examine the difference from *t* to t+1: (22)$$\hat{Q}\left(t\right)=\sum_{n\in N}\sum_{c\in N}{\hat{Q}}_{n}^{c}\left(t\right)$$
(23)$$\Delta t=\sum_{n\in N}\sum_{c\in N}E\left[\frac{{\hat{Q}}_{n}^{c}{\left(t+1\right)}^{2}-{\hat{Q}}_{n}^{c}{\left(t\right)}^{2}}{2\hat{Q}\left(t\right)}\right].$$

The one-slot queue backlog meets
(24)$${\hat{Q}}_{n}^{c}\left(t+1\right)\leq \max\left\{\left({\hat{Q}}_{n}^{c}\left(t\right)-{\hat{X}}_{n}^{c}\right),0\right\}+{\hat{Y}}_{n}^{c}+H\left(A_{n}^{c}\left(t\right)\right),$$
where $X_{n}^{c}\left(t\right)$, $Y_{n}^{c}\left(t\right)$ denotes the packets transmitted along (a,b) and (b,a) at time slot *t*, respectively,

The following inequality,
(25)$${\left(\max\left\{q-b,0\right\}+a\right)}^{2}\leq q^{2}+a^{2}+b^{2}+2q\left(a-b\right),$$
is valid if q≥0,a≥0,b≥0. Thus, the bound of Δt satisfies
(26)$$\Delta t\leq M+\sum_{n\in N}\sum_{c\in N}{\hat{Q}}_{n}^{c}\left(t\right)\times E\left[\left(H\left(A_{n}^{c}\left(t\right)\right)-{\hat{X}}_{n}^{c}\left(t\right)+{\hat{Y}}_{n}^{c}\left(t\right)\right)/\hat{Q}\left(t\right)\right],$$
where *M* and $H(A_{n}^{c}\left(t\right))$ are finite constants. Thus, to minimize the above inequality, we need to maximize
(27)$$E\left[\sum_{n\in N}\sum_{c\in n}{\hat{Q}}_{n}^{c}\left(t\right)\times \left({\hat{X}}_{n}^{c}\left(t\right)-{\hat{Y}}_{n}^{c}\left(t\right)\right)/\hat{Q}\left(t\right)\right],$$
i.e., maximize
(28)$$\sum_{n\in N}\sum_{c\in n}{\hat{Q}}_{n}^{c}\left(t\right)\times \left[{\hat{X}}_{n}^{c}\left(t\right)-{\hat{Y}}_{n}^{c}\left(t\right)\right],$$
where
(29)$${\hat{X}}_{n}^{c}\left(t\right)-{\hat{Y}}_{n}^{c}\left(t\right)=H\left(X_{n}^{c}\left(t\right)\right)-H\left(Y_{n}^{c}\left(t\right)\right)=\sum_{p\in X_{n}^{c}\left(t\right)}H(p)-\sum_{p\in Y_{n}^{c}\left(t\right)}H(p).$$

Let all packets arrive randomly; when the time span is long enough, we can get
(30)$$\frac{H(X_{n}^{c}\left(t\right))}{X_{n}^{c}\left(t\right)}=\frac{H(Y_{n}^{c}\left(t\right))}{Y_{n}^{c}\left(t\right)}=\frac{H(Q_{n}^{c}\left(t\right))}{Q_{n}^{c}\left(t\right)}=\frac{{\hat{Q}}_{n}^{c}\left(t\right)}{Q_{n}^{c}\left(t\right)}.$$

Based on this assumption, we maximize
(31)$$\begin{array}{c} \begin{array}{c} \sum_{n\in N}\sum_{c\in n}{\hat{Q}}_{n}^{c}\left(t\right)\times \left[H\left({\hat{X}}_{n}^{c}\left(t\right)\right)-H\left({\hat{Y}}_{n}^{c}\left(t\right)\right)\right]\\ =\sum_{n\in N}\sum_{c\in n}{\hat{Q}}_{n}^{c}\left(t\right)\times \left[\frac{H\left({\hat{X}}_{n}^{c}\left(t\right)\right)}{Q_{n}^{c}\left(t\right)}X_{n}^{c}\left(t\right)-\frac{H\left({\hat{X}}_{n}^{c}\left(t\right)\right)}{Q_{n}^{c}\left(t\right)}Y_{n}^{c}\left(t\right)\right]\\ =\sum_{n\in N}\sum_{c\in n}\frac{{\hat{Q}}_{n}^{c}\left(t\right)}{Q_{n}^{c}\left(t\right)}\times \left[X_{n}^{c}\left(t\right)H\left(Q_{n}^{c}\left(t\right)\right)-Y_{n}^{c}\left(t\right)H\left(Q_{n}^{c}\left(t\right)\right)\right]\\ =\sum_{n\in N}\sum_{c\in n}\frac{{\hat{Q}}_{n}^{c}\left(t\right)}{Q_{n}^{c}\left(t\right)}\times H\left(Q_{n}^{c}\left(t\right)\right)\times \left[\sum_{b\in N}\mu_{nb}^{c}\left(t\right)-\sum_{a\in N}\mu_{an}^{c}\left(t\right)\right]. \end{array} \end{array}$$

Since ${\hat{Q}}_{n}^{c}\left(t\right)/Q_{n}^{c}\left(t\right)$ is non-negative, the above equality is equivalent to maximizing
(32)$$\begin{array}{c} \sum_{n\in N}\sum_{c\in n}H\left(Q_{n}^{c}\left(t\right)\right)\times \left[\sum_{b\in N}\mu_{nb}^{c}\left(t\right)-\sum_{a\in N}\mu_{an}^{c}\left(t\right)\right]\\ =\sum_{a\in N}\sum_{b\in N}\sum_{c\in n}\mu_{ab}^{c}\left(t\right)\times \left[H\left(Q_{a}^{c}\left(t\right)\right)-H\left(Q_{b}^{c}\left(t\right)\right)\right]\\ =\sum_{a\in N}\sum_{b\in N}\sum_{c\in n}\mu_{ab}^{c}\left(t\right)\times \left[{\hat{Q}}_{a}^{c}\left(t\right)-{\hat{Q}}_{b}^{c}\left(t\right)\right]\\ =\sum_{a\in N}\sum_{b\in N}\sum_{c\in n}\mu_{ab}^{c}\left(t\right)\times {\hat{\omega }}_{ab}^{c}\left(t\right), \end{array}$$
where ${\hat{\omega }}_{ab}^{c}\left(t\right)$ denotes a differential DHD-based backlog of flow *c* of link (a,b). The DHBP routing decision is equivalent to determining $\mu_{ab}^{c}\left(t\right)$ to minimize Δt and maximize Equation (Equation 32). Maximization represents the routing policy of DHBP. Hence, DHBP obtains the optimal throughput by considering the destination hops’ delay in the LEO satellite network.

## 6. Simulation Performance and Analysis

### 6.1. Simulation Setup

The performance of the DHBP routing strategy is evaluated on a giant inclined orbit constellation on the OMNET++-based network simulation platform. The constellation consists of 648 satellites with a total of 18 orbits. In each orbit, 36 satellites are evenly distributed. The orbit has an altitude of 550 km and an inclination of $53^{\circ }$. In terms of network parameter settings [41], we set up four two-way ISLs for each satellite (two inter-plane ISLs and two intra-plane ISLs). The link transmission rates of uplink, downlink, and ISLs are set to 10 Mbps [42]. A constant flow with the constant packet size (512 Byte) is adopted. The time for each round of simulation is set to 100 seconds. The orbital parameters and network parameters are shown in Table 2.

Due to the uneven geographical distribution of global communication load, eight source and destination nodes are set up, which are combined into eight groups of source–destination pairs. The distribution of satellites and traffic nodes is shown in Figure 8. Table 3 lists satellite network traffic pairs. The network traffic is sent from the ground station near the source satellite and is connected to the LSN leveraging the closest satellites. Traffic is transmitted down to the terrestrial network upon reaching the satellite nearest to the destination.

We compare DHBP with distance-based back-pressure routing (DBPR) [4], which uses link distance rather than hops count as the backlog metric of back-pressure routing. In order to ensure the accuracy of the comparison, the permitted transmission region of DHBP and DBPR is limited in a rectangular area. The second comparison routing protocol is open shortest path first (OSPF) [43], which utilizes Dijkstra’s shortest path. The third comparison routing protocol is state-aware and load-balanced routing (SALB) [8]. SALB determines routing paths distributively, and dynamically adjusts the queue delay weight based on link states. The following performance metrics are adopted to comprehensively measure the performance of routing protocols.

Average delay. Average time of transmitting packets from source to destination;Average number of forwarding per packets. The average number of times each packet is forwarded from source to destination, which measures the average delivery cost;Throughput. Overall, the successful delivery rate of the packet to the destination node;Data delivery ratio. The ratio of messages successfully delivered to the destination satellites to all messages sent by source satellites in the network.

### 6.2. Result Discussion

To test the validity of DHBP in the LSN, simulations of DHBP, DBPR [4], OSPF [43], and SALB [8] are carried out under different traffic patterns. The traffic patterns adopted in this paper include constant bit rate traffic, Poisson traffic, and bursty traffic.

#### 6.2.1. Constant Bit Rate Traffic

In constant bit rate (CBR) traffic, the traffic generation rate changes gradually from 2 Mbps to 4 Mbps at the interval of 0.25 Mbps to test the performance of routing protocols under different link loads. The buffer size of each satellite is 50 packets. Simulation results are demonstrated in Figure 9.

The average delay under different routing protocols is shown in Figure 9a. The proposed DHBP uses hops count as the backlog metric. Compared with DBPR, which utilizes the link length as the metric, DHBP reduces the average delay by 14.67% at a traffic rate of 3.5 Mbps. OSPF uses the shortest path based on Dijkstra and only obtains information from neighboring satellites to consolidate routing tables. When the link load is heavy, a large number of packets will be dropped, resulting in the network performance degradation. SALB achieves the highest delay because it postpones data sending upon link congestion. Meanwhile, SALB only select the next hop from no more than two neighbors, which aggravates link congestion.

The average number of forwarding per packets with different protocols is demonstrated in Figure 9b, which measures the transmission cost of each packet. Through restricting the propagation area to a rectangular region between the source and destination of all packets, DHBP has a significant advantage over the comparison protocols. Although the topology of the Manhattan network provides a large number of redundant links for the LSN, we can see that the transmission cost is reduced by restricting transmission within a certain region. The average number of forwarding per packets is significantly less than DBPR, OSPF, and SALB when the CBR rate is less than 3.25 Mbps. When the CBR rate is close to 4 Mbps, DHBP has a similar performance to that of OSPF. Therefore, we can see that, when the traffic is heavy, the proposed scheme still has a low transmission cost.

Figure 9c shows the change of network throughput with traffic generation rate under different routing protocols. As the CBR rate increases, the throughput of several routing protocols shows an upward trend. In comparison, DHBP has the largest network throughput, while OSPF has the smallest. When the traffic generation rate is less than 2.4 Mbps, DHBP and DBPR have a similar throughput. As the link load continues to increase, the throughput of DHBP tends to its maximum of about 24 Mbps. Compared to DBPR, SALB, and OSPF, when the CBR rate is 4 Mbps, the maximum throughout of DHBP is improved by 61.65%, 23.74%, and 10.47%, respectively. We can see that the DHBP can always improve the network throughput, especially when the traffic is heavy.

Figure 9d shows the data delivery ratio with different CBR traffic. When CBR rate advances, the data delivery ratio of DHBP, OSPF, DBPR, and SALB decreases due to heavier transmission load. In comparison, DHBP achieves the best data delivery ratio and load-balancing ability since it considers both back-pressure routing and destination-hops-delay, which balances network traffic without exorbitant cost. When the CBR rate reaches 4 Mbps, the data delivery ratio is improved by 11.59%, 14.93%, and 40% compared with DBPR, SALB, and OSPF, respectively. SALB estimates the link states and updates routing tables through an efficient shortest path tree algorithm between two successive handovers, which shows a higher data delivery ratio than that of OSPF. However, SALB cannot provide adequate available paths to adapt to heavy load. Different from DBPR, which adopts the distance from intermediate satellites to destination satellites as link weights, DHBP utilizes the destination hops delay-based differential backlog between neighboring satellites, which can balance the congestion and delay flexibly. Simulation results show that DHBP achieves the best data delivery ratio, especially when the network traffic becomes heavy.

The quantitative analysis of the network performance of different routing protocols under heavy load (CBR = 4 Mbps) is shown in Table 4. The data in the table are the promotion rate of DHBP compared with OSPF, DBPR, and SALB. We can see that, when the network load is heavy, DHBP has a significant improvement in throughput. Compared with OSPF and SALB, the introduction of the back-pressure mechanism allows for the diversion of network traffic to take into account the load and transmission delay. As a result, both back-pressure-based routing, i.e., DHBP and DBPR, perform better. Compared with DBPR, DHBP has improved delay performance. This is due to the fact that DHBP utilizes destination hops count delay as a backlog metric. In satellite networks, hops count reflects the link delay more accurately than the distance between satellites.

#### 6.2.2. Poisson Traffic

In the real physical world, network traffic can be described by Poisson distribution. In this paper, we test routing protocols under Poisson traffic under various rates. Simulation results are demonstrated in Figure 10. According to the results, similar to CBR traffic, DHBP has the best throughput and the lowest transmission cost. The DDR is higher than that of DBPR and SALB, which indicates a good load-balancing performance.

#### 6.2.3. Bursty Traffic

In LEO satellite networks, the bandwidth occupied by non-real-time services is difficult to predict. Bursty traffic often occurs. Bursty traffic can lead to the degradation of network quality, which can cause congestion and increase forwarding delay. In severe cases, packet loss may occur, resulting in degraded or even unavailable service quality. The performance under bursty traffic is demonstrated in Figure 11.

Different from CBR traffic and Poisson traffic, the transmission period of bursty traffic consists of a bursty phase and a normal phase. The average duration of a bursty phase is 1 second with a random rate change of between 3 Mbps and 11 Mbps. According to Figure 11, performances of DHBP are similar to results under CBR traffic and Poisson traffic. The robustness of DHBP against traffic bursts is verified. Thus, we can conclude that the proposed method outperforms the state-of-the-art routing protocols, with several improvements in transmission cost and throughput.

The average throughput and data delivery ratio under Poisson traffic and bursty traffic are demonstrated in Figure 12. We can see that the proposed scheme outperforms DBPR, SALB, and OSPF in terms of throughput and data delivery ratio. Compared to Poisson traffic, DHBP has more significant advantages under bursty traffic. Since DHBP takes the cache queue length and remaining hops into account, it can divert the bursty traffic to relatively idle nodes within a short period of time, which effectively improves the QoS of the LSN. OSPF and SALB, etc. only consider global congestion without taking the current satellite and neighboring satellites into account. Therefore, it is difficult to make reasonable traffic diversions. The performance deterioration of such routing protocols is more severe when there is a burst of traffic.

## 7. Conclusions

With the properties of wide coverage and high communication capacity, the LSN will be a vital supplement to terrestrial networks and a significant part of SAGIN. Nevertheless, the bursty property and uneven geographical distribution of network traffic result in the load imbalance of the LEO satellite network. To address these issues, we propose distributed hops-based back-pressure routing (DHBP). Firstly, a method for estimating the minimum end-to-end hops count between satellites in inclined constellations is deduced theoretically. On this basis, DHBP takes the queue length of the intermediate satellite and remaining hops count between the intermediate satellite and destination satellites as the backlog metric of back-pressure routing. By dynamically selecting the shortest path to achieve low congestion, the traffic overhead of the satellite network can be balanced. Considering the link redundancy resulting from the property of the Manhattan network, this paper restricts the permitted propagation region to a rectangular area determined by the source and destination nodes, so as to control the transmission cost. Taking an inclined orbit constellation consisting of 648 satellites as an example, the proposed scheme is verified by network simulation. The simulation results demonstrate that, compared with DBPR, DHBP improves performance under CBR traffic in terms of delay, transmission cost, throughput, and data delivery ratio by 12.42%, 13.13%, 10.47%, and 11.59%, respectively. Meanwhile, DHBP has more significant advantages under bursty traffic. Therefore, it is demonstrated that DHBP improves throughput and optimizes the transmission cost, especially under bursty traffic environments.

In our future work, we will further explore more scenarios with a link break and different traffic patterns. In particular, the performance of the LEO satellite network under different link break patterns will be analyzed.

## Figures and Tables

**Figure 1 sensors-23-09789-f001:**
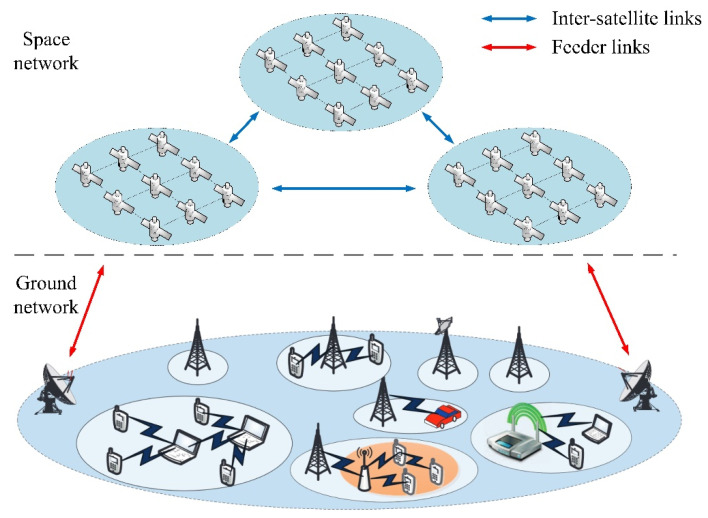
Load-imbalancing problem in LEO satellite network.

**Figure 2 sensors-23-09789-f002:**
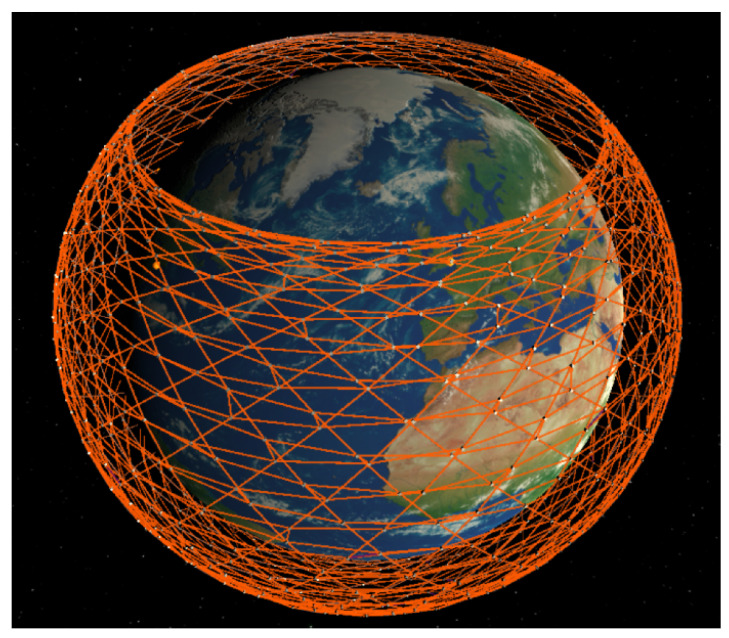
Inter-satellite links of the LEO satellite network.

**Figure 3 sensors-23-09789-f003:**
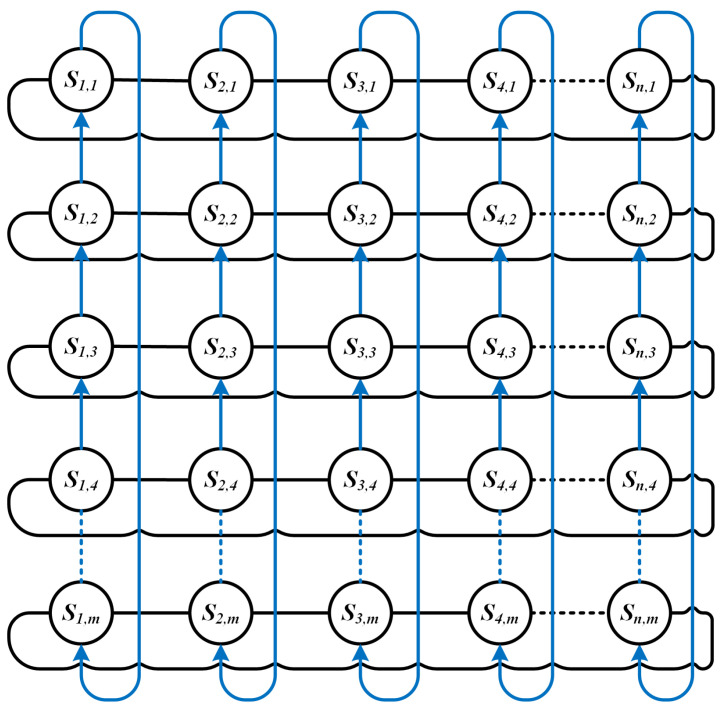
Topology model of the LEO satellite network.

**Figure 4 sensors-23-09789-f004:**
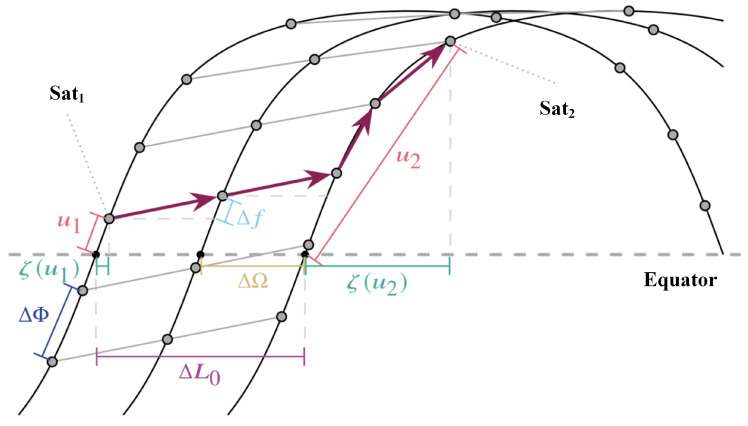
Trajectory of substellar points of LEO satellite constellation.

**Figure 5 sensors-23-09789-f005:**
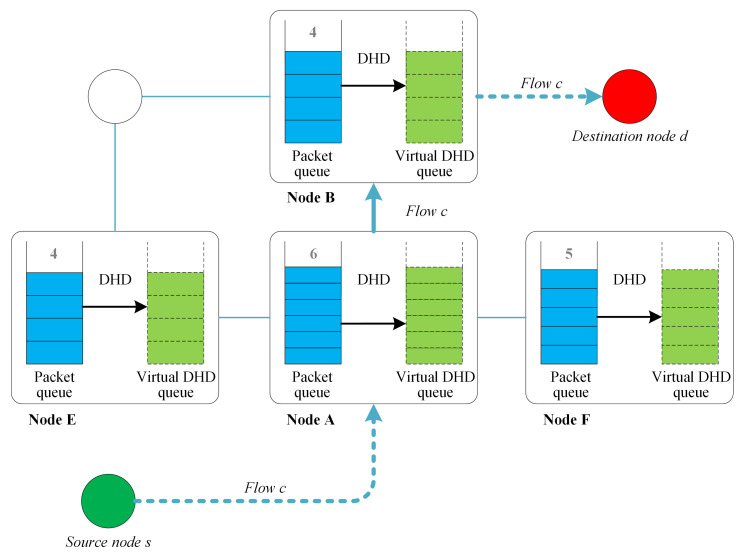
Queue management and backlog calculation of DHBP.

**Figure 6 sensors-23-09789-f006:**
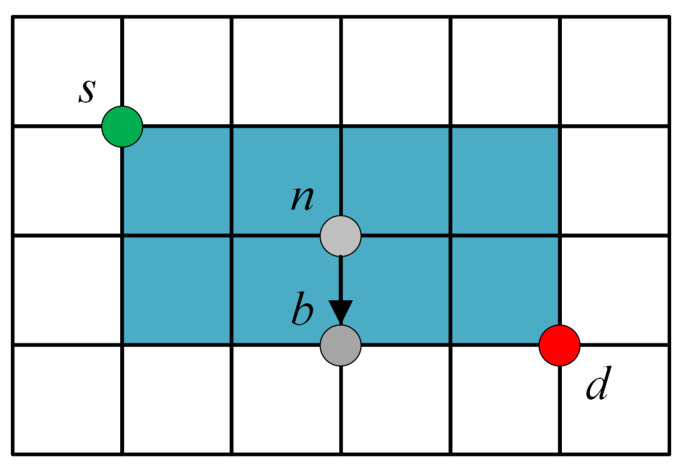
Available rectangular region.

**Figure 7 sensors-23-09789-f007:**
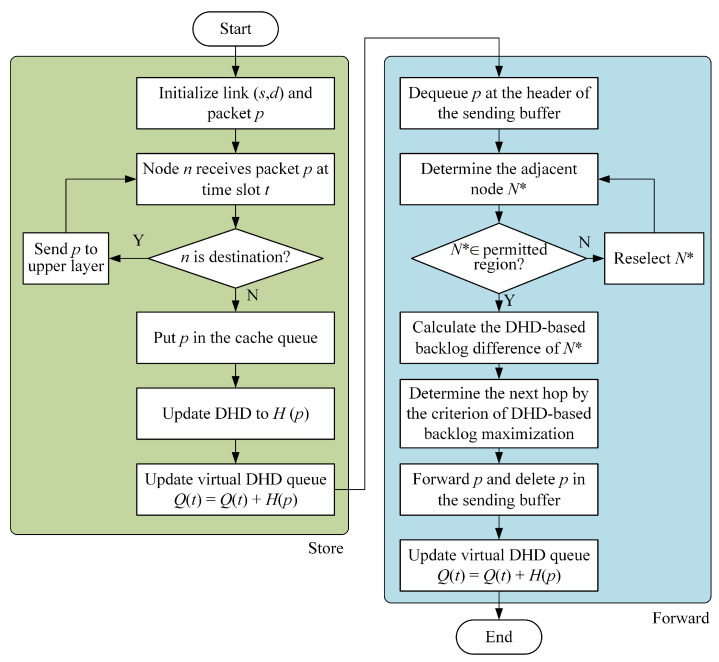
Packet storage and forwarding process of DHBP algorithm.

**Figure 8 sensors-23-09789-f008:**
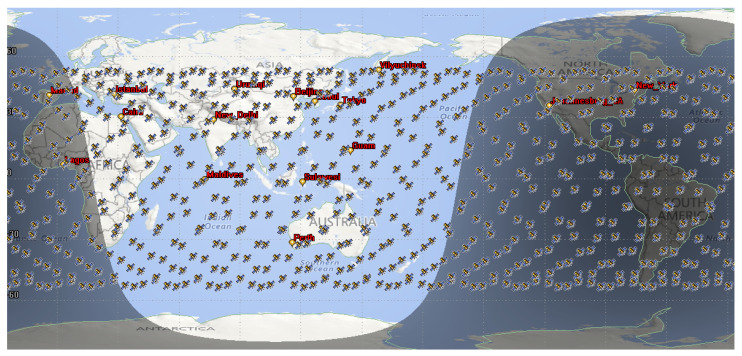
Distribution of LEO satellite constellation and ground nodes.

**Figure 9 sensors-23-09789-f009:**
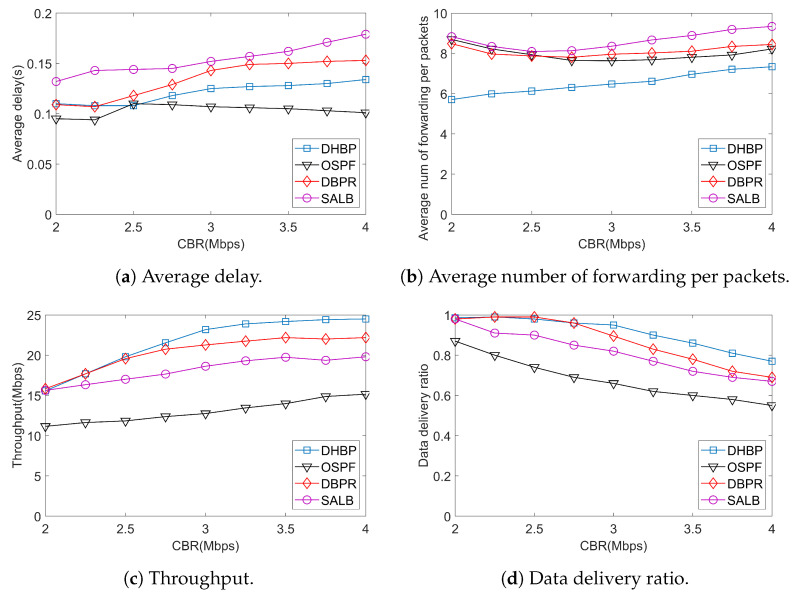
The performance under CBR traffic.

**Figure 10 sensors-23-09789-f010:**
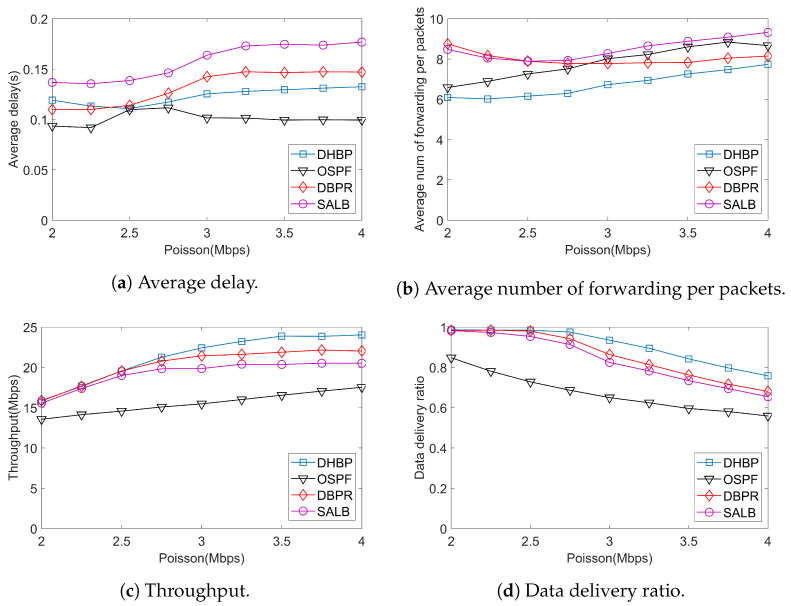
Simulation results under Poisson traffic.

**Figure 11 sensors-23-09789-f011:**
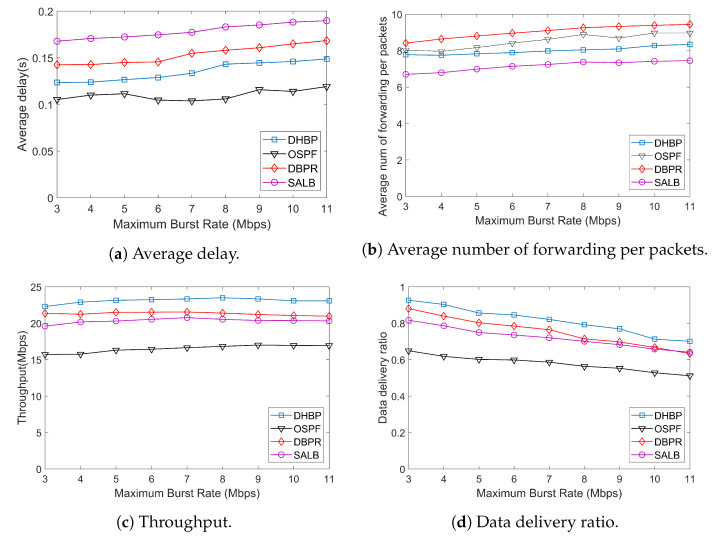
Simulation results under bursty traffic.

**Figure 12 sensors-23-09789-f012:**
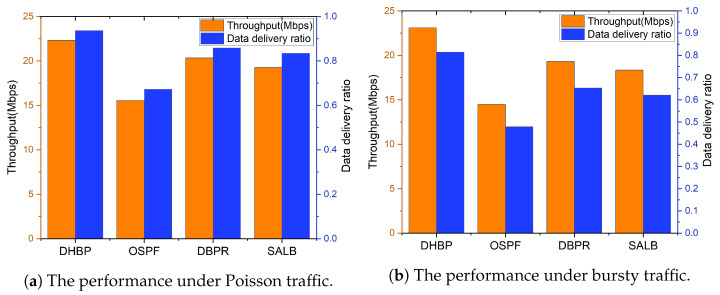
Average performance under Poisson traffic and bursty traffic.

**Table 1 sensors-23-09789-t001:** Comparison of load-balancing routing schemes in the LEO satellite network.

Load-Balancing Method	Scheme	Cost	Overhead Reduction Method
Global load-balancing	LBRA-CP [9] (2019)	Queue delay, geographic location	Fixed/mobile agent
	SLSR [14] (2014)	Propagation delay, queuing delay	Improved flooding algorithm
	SALB [8] (2017)	Queue occupancy	Shortest path tree
	NCMCR [16] (2019)	Propagation delay, queuing delay	No-Stop-Wait ACK
	SIDA [7] (2020)	Link Load	Improved shortest path algorithm
Local load-balancing	TLR [21] (2014)	Propagation delay, queuing delay	Local adjustment
	HGL [23] (2017)	Propagation delay, queuing delay	Local adjustment
	HLBR [30] (2018)	Propagation delay, queuing delay	Local adjustment
	LSP [31] (2019)	Propagation delay, queuing delay	Local adjustment
	DBPR [27] (2023)	Link distance	Restrict rectangular forwarding area

**Table 2 sensors-23-09789-t002:** System parameters.

	Parameter	Value
Orbital parameters	Orbit height (km)	550
	Number of satellites	648
	Number of orbital planes	18
	Number of satellites per plane	36
	Orbital inclination (${}^{\circ }$)	53
Network parameters	Buffer queue size	100
	Minimum elevation of gateway (${}^{\circ }$)	30
	Uplink/downlink rate (Mbps)	10
	Inter-satellite link rate (Mbps)	10
	Packet size (byte)	512

**Table 3 sensors-23-09789-t003:** Satellite network traffic pairs.

Number	Source	Destination
1	Perth	Beijing
2	Madrid	Guam
3	Lagos	Seoul
4	Cairo	Tokyo
5	Istanbul	Sulawesi
6	New Delhi	Johannesburg
7	Urumqi	New York
8	Maldives	Vilyuchinsk

**Table 4 sensors-23-09789-t004:** Quantitative analysis of network performance under heavy traffic load.

Scheme	Average Delay (%)	Average Forwarding (%)	Throughput (%)	Data Delivery Ratio (%)
OSPF [43]	−32.67	10.77	61.65	40.00
DBPR [4]	12.42	13.13	10.47	11.59
SALB [8]	25.14	21.47	23.74	14.93

## Data Availability

The relevant data of this paper can be accessed by contacting 15850466132@163.com.

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
