# Peer review of "Load-Balancing Routing for LEO Satellite Network with Distributed Hops-Based Back-Pressure Strategy"

_sensors, 2023, doi:10.3390/s23249789_

Round 1

Reviewer 1 Report

Comments and Suggestions for Authors

This article proposes an inter-satellite link routing algorithm for low-Earth orbit satellites. This research focuses on the currently popular field of NTN and has a certain degree of research value. The author proposes using a distributed multi-hop BP routing algorithm to address the dynamic changes of satellite nodes, which has a certain degree of innovation. The writing of the paper is relatively standardized and the English expression is good. Overall, the paper is relatively in line with the theme of the special issue, and further modifications could make it eligible for acceptance.

a)     The algorithm used in the article focuses on the optimal routing between a few satellite nodes and can meet local optimality. However, due to the global end-to-end traffic scenario, achieving global optimality is not possible and further solutions are needed.

b)    The article has neglected the inter-satellite link's model, which requires further complementation of the impact of different delays between various satellite nodes, as well as the abstraction and analysis of the probability of link breaks.

c)     The article needs further analysis of the impact of traffic models on routing performance.

d)    The article needs further complementation of relevant work in the past 5 years.

e)     Please provide information on how to predict routing paths based on ephemeris.

Comments on the Quality of English Language

Minor editing of English language required.

Author Response

Thank you very much for taking the time to review this manuscript. Please see the attachment.

Reviewer 2 Report

Comments and Suggestions for Authors

Comments to Author

The authors proposed a distributed hops-based back-pressure routing (DHBP) for estimating the minimum end-to-end propagation hops between any satellite nodes in an inclined orbit satellite constellation is deduced theoretically. On this basis, DHBP takes the remaining hops between the link intermediate node and the destination satellite as the backlog metric of the back pressure routing. By dynamically selecting the shortest path to meet low congestion, the traffic overhead of satellite network can be balanced. The technical part and the methodology that authors used are very good; however, there are serious comments that must be considered in the next revision as follows:

1)      In the proposed method.

·         The authors assumed that the routing based on the minimum hop counts between the source and the destination will balance the traffic load in the satellite networks. However, they have not shown any proof of how that claim can solve the congestion due to the shortest path which will increase the end-to end delay. Furthermore, load balancing means distributing the packet load among the optimal selection of one hop neighbors which means not necessary always the shortest path must be selected. Therefore, I advised the authors to add one or more routing parameters to the hop count such as power consumption, multi-hop forwarding, and predication of congestion.

2)      I suggest changing the title of section 6; "Simulation Results", into "Simulation Performance and Analysis".

3)      The results in section 6 should be enhanced to include the clear the performance analysis with percentage of results and the justification why the proposed mechanism outperforms the baseline protocols. I suggest adding a subsection under section 6 which should be called Results Discussion.

4)      Conclusion should be enhanced to include the performance analysis with percentage of results.

Comments on the Quality of English Language

1)      The repeating of "queue" in line 36-37, " dynamically sets the queue queue delay weight."

2)      The numbers in line 66-68, "adjacent satellites 66 in the same orbit is about 1 972 km. The distance between adjacent satellites in adjacent 67 orbits varies from 880 km to 1 400 km."

3)      The text in the flowchart diagram, especially in decision, is not clear and out of shape.

Author Response

(The authors gave the same response as above.)

Reviewer 3 Report

Comments and Suggestions for Authors

Here are some of my comments:

1) It's unclear what makes this work novel. Please state how new this work is.

2) Could you please elaborate on the cost of link transmission?

3) Please revise the "Introduction" section and include appropriate references.

4) The authors are asked to compare the DHBP with other models found in the literature. 

5) A number of the lines are incomplete and contain typos. Please fix any error that was made. 

Comments on the Quality of English Language

The authors must improve the writing.

Author Response

(The authors gave the same response as above.)

Round 2

Reviewer 3 Report

Comments and Suggestions for Authors

Thanks for addressing my comments  I have no more comments. 

Comments on the Quality of English Language

English must be improved.